# The Role of We-Intention and Self-Motivation in Social Collaboration: Knowledge Sharing in the Digital World

**Darshana Karna and Ilsang Ko ***

College of Business Administration, Chonnam National University, Gwangju 61186, Korea;
darshanakarna@gmail.com
* Correspondence: isko@chonnam.ac.kr; Tel.: +82-625301444

**Abstract:** Governments throughout the world have set social distancing guidelines to manage COVID-19 that reduced opportunities for maintaining social connections through face-to-face interactions. For this study, we conceptualized social collaboration as an intentional social activity in which people are willing to share their knowledge, experience, and expertise. We examined the relative impacts of we-intention (WI), moral trust (MT), and self-motivation (SM) on participation in social collaboration (PSC) and knowledge sharing (KS). We distributed a questionnaire-based survey to a group of Nepalese residents who actively participated in, commented on, and posted questions on social networking sites and received a total of 239 valid questionnaires for analysis. We tested and verified the research model and variables in SPSS 20 to investigate how PSC accelerates KS intention at digital platforms. The standardized path coefficient for PSM to KS was 0.75, suggesting that social collaborator's participation has a strong positive effect on KS purpose. The standardized path coefficients for WI to MT, WI to PSC, WI to SM, MT to PSC, and SM to PSC were 0.55, 0.72, 0.49, 0.42, and 0.67, respectively. All of the values supported the hypothesis and were significant at $p \leq 0.001$.

**Keywords:** COVID-19; digital platform; knowledge sharing; moral trust; social collaboration; we-intention

## 1. Introduction

COVID-19 affected billions of people around the globe, with significant impacts on health, economic, and social domains. Governments worldwide established guidelines for social distancing, quarantine lockdowns, and working from home to contain the global disease outbreak, and although the restrictions for social gathering varied between countries, government policies often involved closing schools and universities, nonessential physical shops, and businesses, as well as limiting public transportation and facilities and all social gatherings. Under these governmental regulations and policies, public face-to-face interaction has decreased significantly, and with limited opportunities to meet in person, people have been using digital platforms to remain socially connected. Even at work during the COVID-19 pandemic, people have maintained social distancing by using digital collaboration tools such as Zoom, Google, Microsoft, Slack, and WebEx.

People connect to collaborate and contribute knowledge via online social networks, the growth and popularity of which are continuously increasing [1]. However, despite the importance of social networks, comparatively little theory-driven empirical research has addressed this new communication and interaction phenomenon. All individuals (and organizations) link to others, by kinship and through work among the main connectors, and there are two distinct types of physical social networks: formal and informal. While formal social networks are mostly conceptualized in organizational contexts, informal social networks are mostly studied in the context of interactions among individuals concerning friendship, common interests, or health issues. Other common places where individuals

connect are schools, places of religious worship, and places where individuals volunteer, and formal and informal networks can easily overlap in such places.

Over the past 25 years, the rapid evolution of communication technology has transformed peoples' ability to communicate and collaborate. Smartphones and rapid technological advances in mobile devices enable people to connect through e-mail, social collaboration sites, news sites, and instant messaging tools that have become essential parts of people's daily lives. Now people can connect 24 h per day on smartphones, tablets, and notebooks in addition to standard computers, and indeed, newer generations are conducting more online business globally through cloud computing and faster Internet connections.

Social networking platforms are virtual communities that allow people to connect and interact online with each other on particular subjects or any subject [2,3]. Membership in online social networks has recently exploded exponentially. Hitwise [4] announced that the market share of the top 20 social networking websites grew by 11.5% just from January to February 2007 and that social networking activity accounted for 6.5% of all Internet traffic in February 2007. As of September 2013, 71% of global Internet users were sharing on social networking sites [5]. Because the objective of online social networks is social interactions and connection, it is appropriate to consider the use of online social networks as collective social action [6–9]. However, because this phenomenon is quite new, there has been little theory-driven empirical research on intentional social actions within online social networks. In this study, we aim to develop and empirically validate a research model on knowledge-sharing (KS) social actions in online social networks.

Specifically, we aimed to confirm how personal values like we-intention (WI), moral trust (MT), and self-motivation (SM) influence an individual to participate in social collaboration driven by KS as the key factor for such collaboration. Bearing these considerations in mind, this paper is structured as follows: The first part presents a brief literature review, followed by a model and hypothesis. We then explain the data collection and analysis and the results of the causal model. Finally, we give a discussion and conclusions including main findings, implications, limitations, and recommendations for future research.

## 2. Literature Review

The COVID-19 situation has heightened digitalization. The pandemic enables a digital transformation to facilitate more and more trends nationally and internationally. Digital technologies are broadly perceived as a promising means to encourage different organizations like healthcare, education, financial, online business, etc. Now, most of the organizations have moved onto digital platforms for delivering services in education, healthcare, businesses, and research, which are surprisingly increasing implementation of digitalization. The COVID-19 crisis has enhanced digitalization processes in all kinds of services in most countries, although at different rates according to their available technology, online facilities, and government policies.

With the current breakneck advances in information technology, social media and social networking systems emerged and expanded rapidly as well. Social media can be characterized as "collective goods produced through computer-mediated collective action" [10], and this collective action leads to the establishment of an implicit social networking structure, which we aimed to further explore. Many social science researchers have investigated social networks [11], but they have generally been limited to rather small systems, and the researchers have often viewed these networks as static graphs whose nodes are individuals and links symbolize various quantifiable social interactions. Recent work in the sociology of knowledge suggests that the ideas one holds to be true are largely functions of the people one interacts with and the authorities a group recognizes [12]. This claim has been demonstrated in small groups [13] and is consistent with the literature on the social production of scientific knowledge [14].

Social collaboration can help an organization break down its knowledge silos and share its explicit knowledge by providing open environments that foster interaction amongst people. The social capabilities of combined platforms make it easier than ever to share

information, ask questions, find answers, and locate experts based on their content. Social collaboration takes the effort out of making knowledge and people discoverable; it enables people to stay current on information they care about, get to know each other, share important information, and connect to organizational systems and services. And meanwhile, just as we follow individuals, we can follow systems or organizations and receive updates about their status and activities. Social connections break down silos and enable connections so that regardless of location or device, people can spend more time working with others and being productive.

Throughout the world, value creation and consumption are shifting from individuals to the collective, organizing structures are moving from closed hierarchies to open networks, task coordination is evolving from top-down to bottom-up, and knowledge transfer is shifting from a linear distribution to dynamic participation. In this new world of social collaboration, organizations are reexamining how they operate and are seeking ways to capitalize on the new efficiencies of agility and robustness. These efficiencies can be gained by sharing knowledge, working together, accelerating learning, and providing connected experiences that empower groups of people to get things done.

*2.1. We-Intention (WI)*

Bagozzi and Lee [15] proposed three levels of explanation for decision-making in social research: (1) classic individual-based models (a personal intention to perform an individual act) [16], (2) contingency, consistency, and other normative-based models (personal intention to perform an individual act but with consideration of the social influence) [17], and (3) group-based models (including both I-intention and WI to perform a group activity) [18]. In the past two decades, research on information systems has been dominated by studies using the classic individual-based models to explain the adoption and initial use of new technology [19]; personal intention to use a new technology depends on the reasons for doing so, including perceived social pressure to use the technology.

Kelman [20] found that social influence and attitude change depending on "compliance, identification, and internalization," and in information systems adoption research, compliance appears to be the dominant process. Second-hand information from family and friends is important in making decisions about new technologies, and primary references are most important when users have no actual user experience with new technologies. In online social networking, users are exposed to more different influences the more they interact in their networks.

Meanwhile, identification refers to the self-awareness of one's membership in a particular group as well as the emotional and evaluative significance of this membership [20], and social identity can build a sense of belonging in online social networking sites when users interpret themselves to be members of the community. Internalization refers to adopting a decision based on the similarity of one's values with the values of other group members; under internalization processes, realizing that one's values or goals are similar to those of other group members should increase one's WI.

WI is defined as the "commitment of an individual to participate in a group to perform a group activity in which the participants perceive themselves as members of the group" [21], and multiple researchers have proposed the concept [21,22]. WI has been considered to express, "we together will perform a joint action." While I-intention is explained by individual-level reasons for performing a personal act [16,23], WI is explained when a person views the self as part of a social representation in performing a group act [15]. WI exists when one believes not only that one can perform one's part of the joint action but that with colleague participants, one can act jointly at least with some non-zero probability [24].

There are many distinguishing features between I-intention and we-intention. The most important one is a target and achieving the process of a goal. The I-intention target is for an individual, and intention content is privately accepted [25]. It is motivated by personal reasons, with full control over a personal activity. Finally, satisfaction conditions are personal. WI involves participants collectively accepting content together and is mainly

motivated by group reasons. The people have a joint commitment and shared authority over collective action in WI. In contrast with I-intention, WI highlights the individual commitment in collectivity and the social nature of group actions. Therefore, WI is a more appropriate perspective for studying online social activities.

### 2.2. Moral Trust (MT)

Trust has become an important topic in disciplines as broad as science, management, ethics, sociology, psychology, and economics. The central idea behind MT is that most people share similar fundamental moral values [26] and that wide ranges of people belong to similar moral communities even as they need not share similar views on policy issues or even similar ideologies. MT holds that although we might have different religious, cultural, and sociological beliefs, people have deeper similarities than differences, and "trust arises when a community shares a set of moral values in such a way as to create regular expectations of consistent and honest behavior" [27].

In a workplace setting, MT holds that employees should show moral character; in any given situation, people should be honest, loyal, and fair and take care of each other's reputations by avoiding sloppy or fraudulent collaborative work [28–31]: Evidence that a potential member A is honest, loyal, and fair and/or cares about his/her peers can provide member B with reason to trust A to avoid damaging B's reputation with bad work. Similarly, evidence of A's good moral character can give B reason to share ideas or materials with A; evidence that A cares about fairness and abhors exploiting people's vulnerabilities can rationalize B's expectation that A will not steal B's ideas or work [32].

Ethical communication is very important for the development of MT in a collaborative environment. It determines how a person uses language, respects, and maintains relationships that are guided by an individual's moral values [33]. It represents being honest, accurate, and truthful. Truthfulness and honesty are vital parameters for ethical communication. In this digital age, accessibility to advance technology is granted to everyone. Therefore, ethical communication plays a vital role in sharing information on digital platforms while connecting and communicating and maintaining MT.

Some authors distinguish MT (which they call simply trust) from mere reliance. Despite the debate on the distinction, some points are clear. In contrast to interactions of mere reliance, relationships based on MT carry moral weight because they possess the possibility of betrayal [34]. As Holton [35] notes, in MT the trustor counts on the trustee's intrinsic moral motivations, whereas the self-interested fear of external punishments is insufficiently internal to be the basis of MT. Thus, the MT phenomenon is trust between social collaborators: members of the community making plans based on the assumption that a group member will do something or care for some valued good.

### 2.3. Self-Motivation (SM)

Humans are social beings and have always banded with each other to solve problems. We work together to achieve the targeted goal. By this, we can limit the gap between each other's abilities, strengthen social bonds, and learn prospects of how problems have been solved. Self-regulation in learning, motivation, and emotion has been widely discussed by several scientists [36–38]. Self-regulated learning is driven by will and regulated by motivation [39].

Motivation means that a person is moved to do something, in contrast with not feeling any impetus or encouragement, or being unmotivated. Motivation differs in not only levels but also kinds, that is, the underlying "why." For example, students might be motivated to learn new skills because they have potential value or will produce positive effects, such as good grades. The amount of motivation does not necessarily vary, but the nature and focus of motivation certainly do. Self-motivation (SM) in particular can be defined as the desire or willingness to share or acquire knowledge because of enthusiasm or interest without needing pressure from others; it affects the nature, strength, and persistence of an individual's behavior.

Wasko and Faraj [40] explained how individual motivation for KS in an electronic network of practice mainly occurs when individuals are motivated to access the network, while SM for social collaboration is driven by pleasure in the KS task itself, existing within individuals absent any external pressure. To extend KS, individuals must think that their contribution to others will be worth the effort and that some new value will be created, with expectations of receiving some of that value for themselves [41]. These personal benefits are more likely to accumulate to individuals who actively participate and help others [42]. Thus, the expectation of personal benefits can motivate individuals to share their knowledge with others in the absence of personal acquaintance, similarity, or the likelihood of direct reciprocity [43].

Kankanhallii et al. [44] explained that self-efficacy relates to people's perception about what they can do with the skills that they have. When people share expertise with group members, they gain confidence in what they can do; this brings the benefit of increased self-efficacy. In the workplace, knowledge self-efficacy typically takes the form of believing that one's knowledge can help solve job-related problems, improve work efficiency, or make a difference to the organization.

### 2.4. Participation in Social Collaboration (PSC)

Social collaboration invites individuals to share and refine divergent viewpoints on the topic of interest to extend their thinking beyond individual capabilities [45]. Social electronic networks link individuals and support them in interacting repeatedly with each other, not only by potentially learning and adapting successful strategies, but even by conditioning their behavior or the behavior of others. Social networks influence and constrain individuals' behaviors in nontrivial ways [46], but they also contribute to aspects generically referred to as social capital [47,48], which favor the emergence of coordinated actions.

PSC involves people at the center of computing that ensures that collaborators, customers, employees, partners, etc., can connect with the right people for the right information. The existing literature sheds light on the role of social ties in all types of networks between individuals [49] or within [50] or across organizations [51–53]. Scholars have studied collaboration networks among scientists and found that they use similar techniques and influence each other's work in precise settings [54] and lab ethnographies [55], but now researchers have devised a social interaction structure that works for all disciplines. Although we might suppose the link between networks and ideas to be strongest in small groups, a logical extension suggests that long-term developments in scientific work might depend on the broader pattern of disciplinary social networks.

Sharing and participating are vitally important for any enterprise that wants to maintain its collective memory, and participation must be simple and rewarding, with embedded social gestures that ebb and flow throughout the systems that people use every day. A platform for social collaboration should be more than just a place where one goes to be social; it should be part of the daily working environment. By providing insightful awareness, gratifying participation, and preserving communal knowledge, social collaboration enhances individuals' experiences and enables them to have embedded cultures of sharing. When social collaboration is deeply embedded in services and applications, the line between consuming and creating blurs, and sharing becomes the new wave.

### 2.5. Knowledge Sharing (KS)

KS is an act through which knowledge (information, skills, or expertise) is exchanged among friends, families, peers, people, and communities, or within or between organizations [56]. KS bonds the individual, and organizational knowledge improves the absorptive and invention capacity, which ultimately leads to the sustained competitive advantage of individuals as well as organizations [57]. Information is considered a flow of messages, whereas knowledge depends on information and is justified by one's beliefs [58]. Many scientists use the terms knowledge and information interchangeably, highlighting that there is not much practical utility in differentiating knowledge from information in KS

research [59,60]. KS refers to delivering information and knowledge to collaborate with others to solve problems, develop new ideas, or implement policies or procedures [61]. KS can happen through written correspondence or face-to-face communications or through documenting, organizing, and capturing knowledge [61].

Previous researchers have already established the use of KS for collaborative work [62,63]. Storck [64] concluded that KS is important for building trust and improving the effectiveness of group work. However, achieving effective KS can involve challenges, particularly in the face of cultural, geographic, or even just time zone differences [65,66]. Faraj and Sproull [67] proposed that instead of sharing specialized knowledge, individuals should focus on knowing where expertise is located and needed, an approach known as transactive memory. Transactive memory is defined as the knowledge group members possess coupled with an awareness of who knows what [68], and researchers have demonstrated that transactive memory positively affected group performance and collaboration by quickly bringing the needed expertise to knowledge seekers [64,67]. Another socially constructed concept is the mechanism that connects individuals and teams to comprise collective knowledge. Grant [69] argues that collective knowledge comprises elements of knowledge that are common to all members of an organization.

Collective knowledge is defined as "a knowledge of the unspoken, of the invisible structure of a situation, certain wisdom" [70], and it can entail profound knowledge of an environment such as established rules, laws, and regulations, or include language or other forms of symbolic communication and shared meaning [69]. Building a sense of collective knowledge in co-located organizations would mean developing a collective mind [71] through participation in tasks and social rituals [72].

## 3. Research Model, Measurement Items, and Hypothesis

### 3.1. Research Model

We developed a research model based on the motivational factors (WI, MT, and SM) that encourage people to participate in social collaboration (PSC) with social collaboration as a mediator between motivational factors and outcome variables knowledge sharing (KS). The proposed model facilitates KS activity through online social collaboration (Figure 1).

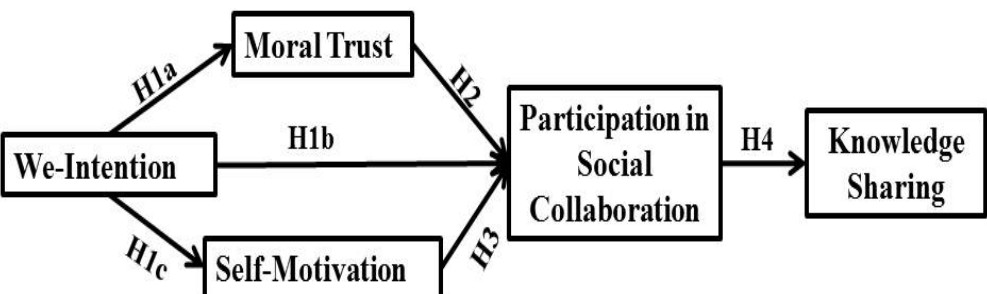

**Figure 1.** A research model for social collaboration in the knowledge-sharing perspective [1].

### 3.2. Measurement Items

KS is an activity in which knowledge transfers from one person, team, or institution to another person, team, or institution, and it depends on open interaction, diversity, and individual creativity [73]. "Open" means that new systems create an open atmosphere for individuals to share knowledge; "interaction" means that more and more people participate and interact with each other to exchange or share knowledge; "diversity" means that participants can share knowledge in many forms, including subscription, evaluation, and recommendation; and "creativity" means user relationships and a growing range of exchange systems that enable innovational and the creation of new knowledge. Knowledge has long been recognized as an important resource for organizational growth and sustained competitive advantage, especially for organizations competing in uncertain environments [74].

Recently, some authors have argued that knowledge is an organization's most important resource because it involves intangible assets and creative processes that are hard to reproduce [69]. However, most organizations do not have all the required knowledge within their official boundaries and must depend on individuals' outside linkages to acquire knowledge [75]. Organizational members benefit from external network connections because they can gain access to new information, expertise, and ideas not otherwise available locally, and they can interact informally, free from the constraints of hierarchy and local rules. KS can save work hours and increase efficiency by increasing the opportunities to learn from past experience and avoiding the repetition of unsuccessful work [76]. Table 1 presents the study measurement variables, their definitions, and how they were operationalized in earlier research efforts.

**Table 1.** Measurement variables, definitions, and prior research efforts.

| Variable | Definition | References |
|---|---|---|
| We-Intention | An individual's commitment to participate in a group to perform group acts in which the participants perceive themselves as members of the group. | [15,21,22] |
| Moral Trust | The moral character of collaborators is reflected as being honest, loyal, and fair and taking care of peers' reputations by avoiding sloppy or fraudulent collaborative work. | [28–30,59] |
| Self-Motivation | Sharing acquired knowledge because of enthusiasm or interest without needing pressure from others. | [40,44] |
| Participation in Social Collaboration | A process that helps people interact and share knowledge, work together, and accelerate learning, providing connected experiences that empower groups of people to achieve common goals. | [46–48] |
| Knowledge Sharing | Sharing information, skills, expertise, and experience among people or communities. | [77,78] |

*3.3. Hypothesis*

3.3.1. WI

WI is a participating intention of a group member to perform an individual part of joint action, together with the other members, while considering that there is a mutual belief (awareness, understanding) about this action [79]. We assumed we-intention as the motivational factor that influences social collaboration behavior [80]. WI focuses on the presence of "we" in intentions for future activities such as the continued use of an online social networking site in the future [81]. WI reflects a cooperative intention among a group of people that everyone will perform their individual roles (having the individual intention to participate in social networks) to perform a joint action with each other (continuing to use the networks together) [81].

**H1a:** *The greater the group WI, the more positive the effect on MT.*

**H1b:** *The greater the group WI, the more positive the effect on SM.*

**H1c:** *The greater the group WI, the more positive the effect on PSC.*

3.3.2. MT

MT is more than mere reliance on the self-interest of one's colleagues. It justifies collaboration with the mere reliance on others' self-interest. Trust in the moral character of individuals provides such reasons for trust. Rather than placing people under strict control mechanisms, group members guided by MT will consider it important for their honor to use their time well [82]. In one study, participants described ideal group members as having moral virtues such as "honesty, accuracy, dependability, loyalty, and cooperativeness" [82]. However, MT is not just a belief; it is also a behavioral disposition. Moral trust, norms, and identification can be considered social capital because they are organizational resources or assets rooted within social relationships that can improve the efficiency of coordinated

action. Therefore, moral trust as evidenced by colleagues' moral virtue can provide reasons for collaboration when mere reliance fails:

**H2:** *The greater the MT, the more positive the effect on social collaboration.*

### 3.3.3. SM

SM for social collaboration has been highlighted as beneficial for individuals to engage in social exchange [83]. When people share knowledge on a social networking site, they might gain self-confidence in terms of what they can do, which increases self-efficacy [84]. This belief can motivate individuals to further contribute and distribute their knowledge [85]. In the work context, knowledge self-efficacy is typically reflected in people's beliefs that their knowledge can help to solve job-related problems [43], improve work efficiency [86], or make a difference to their group [87]. In contrast, people who feel that they lack useful knowledge for social collaboration might not contribute what they know because they believe their knowledge cannot make a positive contribution. In prior research, people who share knowledge in online communities believe in reciprocity [87], a motivational mechanism for people to contribute to open databases [88]. Reciprocity can encourage knowledge contribution because contributors come to expect future help from others in return [88]. Indeed, researchers have observed that people who regularly helped others in virtual communities seemed to receive help more quickly when they asked for it [89], leading to the following hypothesis:

**H3:** *The greater the SM, the more positive the effect on social collaboration.*

### 3.3.4. KS through PSC

When individuals have a common practice of social collaboration, the knowledge readily flows across that practice, allowing individuals to create social networks and support knowledge exchange [90]. Brown and Duguid [90,91] suggest that two types of practice-related social networks are essential for understanding learning, work, and the movement of knowledge: (1) communities of practice and (2) networks of practice. A community of practice involves a tightly knit group of members engaged in a shared practice who know each other and work together, typically meet face-to-face, and continually negotiate, communicate, and coordinate with each other directly. Networks of practice consist of larger, loosely knit, geographically distributed groups of individuals who might be engaged in shared practice, but also might not know each other or even necessarily expect to meet in person [91]. However, participation in social collaboration is open and voluntary, and the people involved are typically strangers. Knowledge seekers have no control over who responds to their questions or the quality of the responses. Knowledge contributors have no assurances that those they are helping will ever return the favor, and lurkers can draw on others' knowledge without contributing anything in return. Thus, we propose the following hypothesis:

**H4:** *The greater the PSC, the more positive the effect on KS.*

## 4. Results

To empirically test the proposed research model, we devised and administered a questionnaire-based survey to social collaborators in Nepal who enthusiastically participated in, commented on, and posted questions on social networking sites. Specifically, we distributed surveys to students at Kathmandu University, Tribhuvan University, and Pokhara University, and of the 250 responses we collected, we discarded 11 because of incomplete responses. Respondents rated the survey items on five-point Likert scales that ranged from 1 = *strongly disagree* to 5 = *strongly agree*, and the items covered motivational factors (WI, MT, and SM), the mediating factor (PSC), and the outcome factor (KS activity

on social networks) (Appendix A). We tested and verified the research model and variables using SPSS 20 AMOS.

Table 2 presents the demographic findings for the study participants. The majority of respondents who were actively participating in social collaboration were aged 40–49 years (41.8%) and had master's degrees (48.5%). By profession, 40.6% were academics, and 62.8% had been involved in social collaboration for more than five years. In addition, 55.2% used social collaboration sites every day, and 36.8% believed that they were using the sites for knowledge sharing.

**Table 2.** The demographic characteristics of the study participants.

| | Category | Frequency | | Category | Frequency |
|---|---|---|---|---|---|
| Age group | 20–29<br>30–39<br>40–49<br>50–59<br>Above 60 | 14 (5.9%)<br>57 (23.8%)<br>100 (41.8%)<br>41 (17.2%)<br>27 (11.3%) | Profession | Students<br>Academics<br>Business<br>Self-employed | 71 (29.7%)<br>97 (40.6%)<br>53 (22.2%)<br>18 (7.5%) |
| Involvement in social collaboration | Less than 1 year<br>1–5 years<br>More than 5 years | 14 (5.9%)<br>75 (31.4%)<br>150 (62.8%) | Qualification | Undergraduate<br>Postgraduate<br>Doctorate | 69 (28.9%)<br>116 (48.5%)<br>54 (22.6%) |
| Frequency of using social collaboration network | Every day<br>Twice a week<br>Once a week<br>Once a month | 132 (55.2%)<br>21 (8.8%)<br>57 (23.8%)<br>29 (12.2%) | Purpose of using social collaboration network | Posting questions<br>Posting answers<br>Seeking knowledge<br>Knowledge contribution<br>Experience sharing<br>Checking for friends' updates | 25 (10.5%)<br>34 (14.2%)<br>32 (13.4%)<br>88 (36.8%)<br>35 (14.6%)<br>25 (10.5%) |

Five multiple-item constructs were subjected to multiple regression analysis, and we tested the constructs for uni-dimensionality, convergent validity, internal consistency, and discriminant validity. We also conducted exploratory factor analysis, and all factor loadings were significant at $p < 0.01$. To assess the reliability and convergent validity of our model, we checked Cronbach's $\alpha$, squared multiple correlations, construct reliability, and average variance extracted (AVE). Cronbach's $\alpha$ estimates the proportion of the variance in a test score that can be attributed to true score variance, and it is used to estimate the proportion of the variance in a score that is systematic or consistent within a set of scores. AVE for each construct was more than 0.800, demonstrating high consistency [92], and Cronbach's $\alpha$ for the constructs ranged from 0.803 to 0.907, which also demonstrated high consistency. All AVEs were greater than the 0.5 cutoffs, indicating satisfactory convergence validity. We measured discriminant validity following Chain's method [93], and the square root of the AVE for each construct was greater than the correlation of the construct with any other constructs, which confirmed the discriminant validity of the measurement model. Table 3 presents all these variables and indicates that all were significant at 0.01.

**Table 3.** The results of the measurement model assessment and constructs correlations.

| Constructs | Cronbach's $\alpha$ | Construct Reliability | SMC | AVE | Constructs Correlation | | | | |
|---|---|---|---|---|---|---|---|---|---|
| | | | | | WI | MT | SM | PSC | KS |
| WI | 0.803 | 0.789 | 0.662 | 0.632 | 0.795 [+] | | | | |
| MT | 0.907 | 0.822 | 0.673 | 0.635 | 0.772 | 0.799 [+] | | | |
| SM | 0.899 | 0.799 | 0.604 | 0.657 | 0.712 | 0.722 | 0.812 [+] | | |
| PSC | 0.891 | 0.801 | 0.741 | 0.667 | 0.633 | 0.641 | 0.742 | 0.821 [+] | |
| KS | 0.902 | 0.799 | 0.681 | 0.682 | 0.628 | 0.638 | 0.667 | 0.750 | 0.826 [+] |

SMC = squared multiple correlation, WI = We-Intention, MT = Moral Trust, SM = Self-Motivation, PSC = Participation in Social Collaboration, KS = knowledge Sharing, [+] = $\sqrt{AVE}$, and all values were significant at $p < 0.01$.

We evaluated common method variance by conducting confirmatory factor analysis, comparing the five-factor model with a single-factor model (or Harman's one-factor model) in which all the indicators were loaded onto a single factor (Table 4) [94,95]. According to Podsakoff et al., if a common method of variance is substantial, then the single-factor model provides a better fit. However, in this study, the single-factor model did not provide a good fit with the data ($\chi^2 = 987$, df = 199, GFI = 0.59, CFI = 0.79, and RMSEA = 0.28), indicating that common method bias was not a serious problem [94,96].

**Table 4.** A Rotated Factor Matrix.

| Constructs Items | Communalities | Factor | | | | |
|---|---|---|---|---|---|---|
| | | 1 | 2 | 3 | 4 | 5 |
| WI2 | 0.508 | 0.706 | 0.086 | 0.051 | 0.155 | 0.008 |
| WI1 | 0.653 | 0.671 | 0.077 | 0.154 | 0.177 | 0.298 |
| WI4 | 0.790 | 0.637 | 0.238 | 0.062 | 0.124 | −0.091 |
| WI3 | 0.533 | 0.621 | 0.344 | 0.123 | 0.024 | 0.076 |
| MT1 | 0.550 | −0.124 | 0.701 | 0.122 | 0.037 | 0.122 |
| MT4 | 0.633 | −0.069 | 0.666 | −0.054 | 0.114 | −0.054 |
| MT2 | 0.567 | 0.012 | 0.641 | 0.099 | 0.102 | −0.055 |
| MT3 | 0.473 | 0.098 | 0.599 | 0.024 | 0.186 | −0.027 |
| SM2 | 0.569 | 0.100 | 0.160 | 0.687 | 0.073 | 0.232 |
| SM3 | 0.389 | 0.166 | 0.079 | 0.642 | 0.141 | −0.082 |
| SM1 | 0.567 | 219 | 0.099 | 0.583 | 0.241 | −0.078 |
| SM4 | 0.546 | −0.068 | 0.048 | 0.511 | 0.067 | 0.175 |
| PSC3 | 0.851 | 0.120 | 0.094 | 0.118 | 0.682 | 0.170 |
| PSC1 | 0.589 | 0.164 | 0.069 | 0.032 | 0.675 | 0.174 |
| PSC2 | 0.726 | 0.169 | 0.099 | 0.035 | 0.634 | 0.184 |
| PSC4 | 0.508 | −0.124 | 0.039 | 0.070 | 0.579 | −0.013 |
| KS2 | 488 | 0.308 | 0.222 | 0.111 | 0.031 | 0.679 |
| KS1 | 0.581 | 0.100 | 0.134 | 0.143 | 0.073 | 0.645 |
| KS3 | 0.404 | 0.166 | 0.038 | 0.062 | 0.124 | 0.628 |
| KS4 | 0.400 | −0.068 | 0.101 | 0.131 | 0.183 | 0.622 |
| **Total Value** | | **8.316** | **1.359** | **1.050** | **0.847** | **0.658** |
| **Variance %** | | **34.649** | **5.662** | **4.373** | **4.527** | **3.740** |
| **Cumulative %** | | **34.649** | **40.311** | **44.684** | **49.221** | **52.951** |
| **Survey Questions** | | **4** | **4** | **4** | **4** | **4** |

Extraction methods: alpha factoring; rotation method: varimax with Kaiser normalization.

We examined the structural model including the research path of dependent and independent variables using AMOS (Figure 2). Path analysis is a statistical technique used to inspect the comparative strength of direct and indirect relationships among variables by estimating a series of parameters by solving one or more structural equations to test the fit of the data. It is useful in making the rationale of conventional regression calculations unambiguous [97].

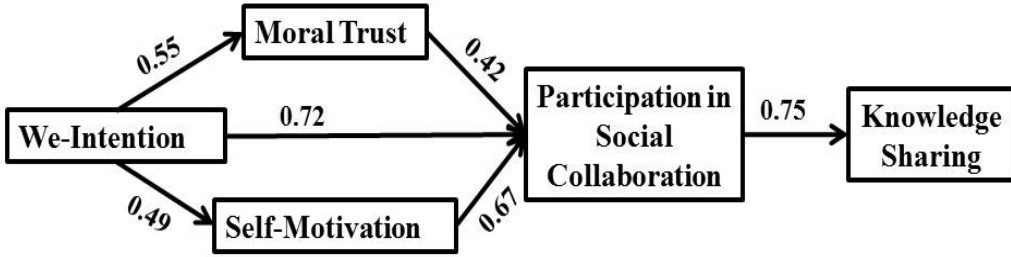

**Figure 2.** The path analysis results of the proposed model.

The proposed five-factor model provided a good fit for the data. Figure 2 details the path analysis with statistically significant values [98]. Table 5 summarizes the hypothesis testing, providing a standardized path coefficient and the results of the statistical tests.

These results provide support for our proposed hypotheses. Specifically, the standardized path coefficients for WI to PSC, WI to SM, MT to PSC, and PSC to KS were 0.72, 0.49, 0.42, and 0.75, respectively. All the values support the hypotheses with significance at $p \leq 0.001$.

**Table 5.** A Summary of Hypothesis Testing.

| Hypothesis/Path | SPC | *p*-Value | Results | Fit Indices |
|---|---|---|---|---|
| H1a: WI → MT | 0.55 | *** | Supported | $\chi^2$ = 214.37 |
| H1b: WI → PSC | 0.72 | *** | Supported | $p$ = 0.000 |
| H1c: WI → SM | 0.49 | 0.001 | Supported | df = 109 |
| H2: MT → PSC | 0.42 | *** | Supported | GFI = 0.91 |
| H3: SM → PSC | 0.67 | 0.001 | Supported | AGFI = 0.87 |
| | | | | RMR = 0.04 |
| H4: PSC → KS | 0.75 | *** | Supported | RMSEA = 0.07 |
| | | | | CFI = 0.93 |
| | | | | NFI = 0.90 |

SPC: Standardized path coefficient, *** = significant at <0.001.

## 5. Discussion

Due to the spread of COVID-19, there has been a huge adjustment in the life of human beings. People think digital platforms are the best alternatives for them to be attached to each other. Digital technology not only plays an important role, but also specifies and resolves many problems during the COVID-19 pandemic. People take into consideration the use of more digital media to keep in touch with their superiors, subordinates, collaborators, family, and friends. It has a strong implication in the lives of people around the world and pledges the admittance to online digital platforms for various activities such as conferences, seminars, meetings, social networking, e-commerce, healthcare, entertainment, and accessing all types of information that is available online. Technology has shown an advantageous and requisite tool to support a disaster.

With this study, we investigated how participation in social collaboration accelerates knowledge sharing on digital platforms. We studied the effects of we-intention, moral trust, and self-motivation on PSC and found that PSC facilitated KS activities. The standardized path coefficient for WI to MT indicated that social collaborators' WI had strong positive effects on individual MT for KS purposes. In social collaboration, group members share and exchange knowledge, skills, and experience with each other, and we believe that MT, both directly and indirectly, supports team members in having the faith to participate in collaborative environments. The standardized path coefficient for SM to PSC indicated that SM was essential for participating in social collaboration for KS. All of the remaining path coefficients demonstrated that social collaborators intend to participate on social network sites to share their knowledge, skills, and experience.

Human beings have an instinctive need to connect, communicate, and collaborate. Digitalization has uncovered this true nature of humans and that truth changes everything. The nearly worldwide available broadband and wireless connectivity mean that there are no limits on social networking locations. By 2015, five billion people were connected through mobile devices [10], and as of now, societies are moving unavoidably toward being mobile societies, with our mobile devices becoming the remote controls of our daily lives. With the advancement of digital mobile communication, like 4G technology, users can freely use the internet to communicate and connect all over the world at video level from their devices. As a consequence of the long-term COVID-19 environment, people feel it is more convenient to use mobile phones to perform each and every activity of daily life, such as participation in social platforms, shopping, entertainment, education, etc. Because of COVID-19 and the advances in technology and connectivity, individuals create billions of social interactions and social transactions every day.

Individual factors refer to personal traits that affect whether or not social ties are created. Throughout life, people add friends because of their specific contact needs, and age as an individual characteristic is closely connected to influence; the choice of sex has

specific ties over time as well. Moreover, common membership within a group, such as on sports teams or community groups, in the same class or year at college, or online discussion groups, can induce feelings of identification and being similar. The main purpose of this study was to examine and analyze the major factors that influence KS activity in social collaboration, and we presented a structured outcomes model of social collaboration in the form of factors and sub-factors. KS in networks of practice is a socially complex process that involves a variety of participants with different needs and goals. In social networks, individuals share their knowledge and help others despite the lack of face-to-face relationships and the fact that others might use their knowledge and never reciprocate. In social collaboration, however, group members are motivated by both self-interest and moral regard for their group; they are motivated by both self-interested desires for credit and reputation and moral virtue and a distaste for taking advantage of others.

People share knowledge in social networks because they enjoy helping others; they contribute when they are structurally embedded in networks and when they have the experience to share with others. Self-interested collaborators know that many of their colleagues have moral virtues; accordingly, they do not rely solely on their own and other group members' self-interest to avoid the risks inherent in collaboration, but also often morally trust each other. This MT is particularly salient in situations when powerlessness and self-defeating detection make it irrational to merely rely on one's colleagues. All fundamental political problems are problems of relationships; therefore, all fundamental solutions have to involve fundamental changes in relationships [11]. Commitment to collaboration not only redefines our relationships with each other, but supports creating relationships with the wider world [12].

The purpose of this research is to investigate the association between WI, MT, SM, and PSC on KS perspective. KS plays a major role in the growth of societies. Academics, scientists, researchers, and collaborators have been involved in the study in KS for the advancement of society. Our research result established that PSC accelerates KS on digital platforms that enable sustainability for the sharing of knowledge behaviors. As we know, knowledge is an essential source for sustainability and sharing it plays a crucial role in gaining and making a competitive advantage.

## 6. Conclusions

The COVID-19 pandemic dramatically pushed research online and initiated a new era of digital collaboration; most of the research work that has taken place since early 2020 has been on some kind of digital platform. Research today already involves significant collaboration across national, international, and even intra-institutional settings, so researchers were usually able to quickly move online, and COVID-19 did not have serious impacts on much research. Social collaboration keeps people at the center of computing and ensures that individuals connect with the right people and information, and this did not change in academia.

Participation and sharing are extremely important for successful collaboration because they facilitate the brainstorming processes by which new concepts arise through individual contributions. Online communication platforms are how companies can "go social," and some type of platform should be part of daily working environments. These tools help individuals share views and ideas beyond their professions. The role of knowledge sharing plays a vital role during the pandemic. The present study identifies knowledge sharing as a key active coping behavior that can shape the effects of COVID-19 and encourages academics and collaborators to contribute more to the social network.

## 7. Limitations

Although this study provides insights for KS in social collaboration, it nevertheless has several limitations. We focused on broadening the known antecedents of individual motivation and their impacts on participation in social collaboration, but we believe that additional factors we did not study also have impacts on the attitudes or behavioral

intention of social collaborators to participate in online social networks. For instance, we did not consider privacy in this research, and privacy concerns have significant impacts on individuals' social networking activity. Additionally, we conducted cross-sectional data collection only in Nepal and only among university students and employees. Thus, it is likely that these findings cannot be easily generalized to social collaboration dynamics in other countries because of differences in culture, education, and society between Nepal and other nations. Most of the participants were from an educated background, therefore it could not be generalized within the country. Lastly, the speed of the internet also plays an important role in participation in social collaboration.

## 8. Recommendations for Future Research

This research suggests several important future directions for scholars. First, this research model needs to be further validated and improved concerning both the theoretical basis for the model and its empirical applicability. Future researchers could also add more variables, such as social trust and behavioral intention, because these might also be relevant in participation in social collaboration through knowledge sharing. Other factors could motivate people to participate in social networks and maintain online social network ties, and including new variables in the research model could reveal additional relationships that need to be examined theoretically and empirically. Second, as noted above, future researchers could expand the generalizability or applicability of these findings by applying the model in larger and more varied samples to broader theoretical and empirical examinations of intentional social actions in online social networks.

**Author Contributions:** Conceptualization, D.K. and I.K.; methodology, D.K. and I.K.; data curation, D.K.; writing—original draft preparation, D.K.; writing—review and editing, I.K.; visualization, D.K.; supervision, I.K. All authors have read and agreed to the published version of the manuscript.

**Funding:** This research received no external funding.

**Institutional Review Board Statement:** Not applicable.

**Informed Consent Statement:** Not applicable.

**Data Availability Statement:** Not applicable.

**Conflicts of Interest:** The authors declare no conflict of interest.

## Abbreviations

The following abbreviations are used in this manuscript:

| | |
|---|---|
| AMOS | Analysis of a Moment Structures |
| AVE | Average Variance Extracted |
| CFI | Comparative Fit Index |
| GFI | Goodness of Fit Index |
| KS | Knowledge Sharing |
| MT | Moral Trust |
| PSC | Participation in Social Collaboration |
| RMSEA | Root Mean Square Error of Approximation |
| SM | Self-Motivation |
| SMC | Squared Multiple Correlation |
| SPC | Standardized Path Coefficient |
| WI | We-Intention |

## Appendix A

**Table A1.** Constructs and Measurement Items [1].

| Constructs | Items |
| --- | --- |
| WI | 1. I intend for our group (i.e., the group I identified before) to interact together sometime during the next two weeks.<br>2. I intend for our group to suggest a solution to my problem after I post my question.<br>3. We (i.e., the group I identified) will interact together sometime during the next two weeks.<br>4. We intend for our group to share knowledge among members. |
| MT | 1. I feel that participation will improve my status in the group.<br>2. I earn respect from group members after participating in group activities.<br>3. I trust that someone in the group would help me if I were in a similar situation.<br>4. I have confidence in my ability to provide knowledge that group members consider valuable. |
| SM | 1. I enjoy sharing my knowledge in the group.<br>2. I like to help group members.<br>3. I know that other members of the group will help me, so it is fair to help them.<br>4. It feels good to help group members to solve their problems. |
| PSC | 1. People who are important to me think that I should participate in social collaboration.<br>2. People who influence my behavior think that I should participate in social collaboration.<br>3. I am engaged in this activity to contribute to the pool of information.<br>4. I am engaged in this activity to contribute my knowledge. |
| KS | 1. I believe that group members are willing to share the best knowledge with each other.<br>2. I believe that knowledge contributed by group members is beneficial for my queries.<br>3. I believe that knowledge shared by group members is accurate.<br>4. I believe that group members are honest in KS. |

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
