# Peer review of "The Role of We-Intention and Self-Motivation in Social Collaboration: Knowledge Sharing in the Digital World"

_sustainability, doi:10.3390/su14042042_

Round 1

Reviewer 1 Report

The article is aimed to study to confirm how personal values like we-intention, moral trust, and self-motivation influence an individual to participate in social collaboration driven by knowledge sharing as the key factor for such collaboration.

The research topic is interesting and relevant. The abstract reflects the content of the article.
However, this article needs a profound revision of the theoretical basis.
The article is submitted to the journal Sustainability. The authors did not prescribe how the solution to their issue would contribute to sustainability. The article's content is suitable for the topic of Sustainability, but the authors must show it.

The introduction of abbreviations is not justified. Moreover, abbreviations do not improve the accessibility of communicating the author's thoughts to the reader. The reader does not have to write abbreviations to understand the text. And such a need arises.

Authors touch on very significant concepts without the proper presentation. Section 2.1 We-Intention (WI) (lines 114-148) does not adequately provide theoretical positions on this issue. Thus, there is no discussion of the interrelated concepts of intention and intentionality.

Significant works are the works of E. Husserl and his followers. Also, the author should refer to the works:

Wilson, G., 1989, The Intentionality of Human Action, Stanford, CA: Stanford University Press.

Ginet, C., 1990, On Action, Cambridge: Cambridge University Press.

I-intention and We-intention also include the causal relationship between action and intention. For proper coverage of this issue, the authors should refer to the following works:

Anscombe, G. E. M., 1963, Intention, second edition, Oxford: Blackwell. Anscombe, G. E. M., 1983, 'The Causation of Action,' reprinted in Human Life, Action, and Ethics, M. Geach and L. Gormally (eds.), Exeter: Imprint Academic, 2005, pp. 89-108. Anscombe, G. E. M., 1989, 'Practical Inference,' reprinted in Human Life, Action, and Ethics, M. Geach and L. Gormally (eds.), Exeter: Imprint Academic, 2005, pp. 109-147. Archer, A., 2017, 'Do We Need Partial Intentions?' Philosophia 45: 995-1005. Campbell, L., 2018, 'Two Notions of Intentional Action? Solving a Puzzle in Anscombe's Intention, 'British Journal for the History of Philosophy, 26: 578-602.

Of course, these works are just the beginning to get acquainted with this issue.

The authors interpret the issue of Moral Trust (Clause 2.2.) as a communicative phenomenon. However, there remains a hanging question of how moral requirements are formed in communication in the direct collective experience of human relations.

Authors should refer to the following works: Rationalism in politics, and other essays by Oakeshott, Michael, 1901-1990 Luhmann N. Trust and Power. Chichester, John Wiley and Sons Inc., 1979

Of course, the authors have more potential and can conduct further research on this issue.

 2.3. Self-Motivation. There are no interpretations of this concept in various approaches. The disclosure of this term is present in deep research in psychology, pedagogy, sociology, philosophy.

Authors should show that they are familiar with these traditions and indicate their position. Theoretical research should be a critical analysis of the literature. However, authors can also choose the way of showing traditions and approaches, indicating which tradition they follow. At this point, the theoretical analysis of the literature looks very shallow and requires improvement.

Line 270 - 3.2 Hypotheses This section does not present hypotheses that appear later in the text.

The Conclusion does not present conclusions.

Line 569 - twice 33.33 м

Author Response

 We improved our paper according to reviewer's comments and suggestions.

S. No.

Issue

Improvement

1

The solution to their issue would contribute to sustainability

Improved in line number 511-517

2

Abbreviations justified

Improved in line number 569-582

3

Section 2.1 We-Intention (WI) (lines 114-148) does not adequately provide theoretical positions on this issue. Thus, there is no discussion of the interrelated concepts of intention and intentionality.

Improved and proper cite with reviewer suggestion. Improved in line number 156-162

4

Moral Trust (Clause 2.2.) as a communicative phenomenon. However, there remains a hanging question of how moral requirements are formed in communication in the direct collective experience of human relations.

Improved and proper cite with reviewer suggestion. Improved in line number 183-189

5

 2.3. Self-Motivation. There are no interpretations of this concept in various approaches. The disclosure of this term is present in deep research in psychology, pedagogy, sociology, philosophy

Improved according to reviewer suggestion in line number 200-205

6

Line 270 - 3.2 Hypotheses This section does not present hypotheses that appear later in the text.

Improved in line number 325

7

The Conclusion does not present conclusions

Modified in line number 532-535

S. No.

Issue

Improvement

1

The solution to their issue would contribute to sustainability

Improved on line number 511-517

2

Abbreviations justified

Improved in line number 569-582

3

Section 2.1 We-Intention (WI) (lines 114-148) does not adequately provide theoretical positions on this issue. Thus, there is no discussion of the interrelated concepts of intention and intentionality.

Improved and proper cite with reviewer suggestion. Improved in line number 156-162

4

Moral Trust (Clause 2.2.) as a communicative phenomenon. However, there remains a hanging question of how moral requirements are formed in communication in the direct collective experience of human relations.

Improved and proper cite with reviewer suggestion. Improved in line number 183-189

5

 2.3. Self-Motivation. There are no interpretations of this concept in various approaches. The disclosure of this term is present in deep research in psychology, pedagogy, sociology, philosophy

Improved according to reviewer suggestion on line number 200-205

6

Line 270 - 3.2 Hypotheses This section does not present hypotheses that appear later in the text.

Improved in line number 325

7

The Conclusion does not present conclusions

Modified in line number 532-535

Reviewer 2 Report

The manuscript "The Role of We-Intention and Self-Motivation in Social Collaboration: Knowledge Sharing in the Digital World" is similar to the article, [D. Karna and I. Ko, "We-Intention, Moral Trust, and Self-Motivation on Accelerating Knowledge Sharing in Social Collaboration," 2015 48th Hawaii International Conference on System Sciences, 2015, pp. 264-273, doi: 10.1109/HICSS.2015.40.] This paper, Karna & Ko, 2015, is not cited, nor referenced in the bibliography.

So it would be interesting if the autors could incorporate in the bibliography the 2015 article, improve the wording to eliminate the textual phrases and specify the contributions of this new work, since it seems that both articles are based on the same empirical research (Line 356ff) and (2015,268ff).

Author Response

We improved our paper according to the reviewer's comments and suggestions.

S. No.

Issue

Improvement

1

If the authors could incorporate in the bibliography the 2015 article, improve the wording to eliminate the textual phrases and specify the contributions of this new work, since it seems that both articles are based on the same empirical research (Line 356ff) and (2015,268ff).

Citied our HICSS article

Reviewer 3 Report

Using a structural model the authors in this manuscript examine the relationships between WI, MT, SM, PSC, and KS. The statistics support the hypotheses that the authors have proposed. I have a few questions and concerns that I  would like the authors to comment on.

1. It will be helpful to give some idea of the raw data used in this study. The author can perhaps provide a descriptive table. What remains unclear to me is how the variables were constructed from the response to the questionnaire.

2. This is related to the specification of the structural model: did the authors explicitly include the co-variance among the exogenous variables? 

3. Were any alternative models based on a different set of hypotheses considered for comparison? (e.g. a direct relationship between WI and KS or MT and KS)

4. Some limitations of the study are outlined by the authors (Sec. 7) especially it not being generalizable to other countries. Looking at Table 2, it seems that most of the participants are young and are highly educated. Thus I feel like it may not be generalizable to population within the country. Other factors, such as access to Internet could also play a role and limit the conclusions.

Author Response

1.      The research design, questions, hypotheses and methods can be improved.

Improvement was made on line number 296-297 and 329-31.

Issues 1. It will be helpful to give some idea of the raw data used in this study. The author can perhaps provide a descriptive table. What remains unclear to me is how the variables were constructed from the response to the questionnaire.

We provide Table 1. Descriptive Statistics.

Issue 2. This is related to the specification of the structural model: did the authors explicitly include the co-variance among the exogenous variables? 

Please see Table 3. Correlation Coefficients.

Issue 3. Were any alternative models based on a different set of hypotheses considered for comparison? (e.g. a direct relationship between WI and KS or MT and KS)

We consider WI, MT and SM as motivational factor followed by PSC as mediating factor and KS as outcome factor. We think there is not direct relationship between WI and KS or MT and KS because of the presence of mediating factor.

Issue 4. Some limitations of the study are outlined by the authors (Sec. 7) especially it not being generalizable to other countries. Looking at Table 2, it seems that most of the participants are young and are highly educated. Thus I feel like it may not be generalizable to population within the country. Other factors, such as access to Internet could also play a role and limit the conclusions.

Improvement was made on limitation section, line number 556-558.

Reviewer 4 Report

Thank you to the authors for this paper, I was very keen to read it. The research provided is very interesting, although it has many issues that I will elaborate now: 

The paper "The Role of We-Intention and Self-Motivation in Social Collab3 oration: Knowledge Sharing in the Digital World" starts with COVID19 impact on social interaction, however in discussion part it does not refer on COVID19 at all. This is reasonable since the authors did not have in the research  Constructs and measurement items any item related to COVID19. The more concernable is the fact that the paper is related to Digital world and the research construct does not contain any item on digital technologies.

The authors discuss knowledge sharing, and in some sections of the paper mention working environment. It is not clearly stated if research is oriented in business environment.  In any case, the problem statement should be more clearly defined. The literature review should be more profound, since there are many research papers on social network in digital environment, on the impact of knowledge sharing on business performances etc. 

On my opinion, the scientific contribution in this paper is missing, literature review needs to be improved, problem statement improved, research construct should involved digital technologies, and the results should be related to existing literature and the context of the problem, to finally stated the contribution of the paper. 

I encourage author to continue with their work, still the paper in this form on my opinion is not for the publication in this journal.

Author Response

We improved our paper according to the reviewer's comments and suggestions.

S. No.

Issue

Improvement

1

COVID part in discussion

Improved in line number 456-465 and line number 511-517

2

Measured item related to COVID

We have focused our research in participation in social collaboration and because of COVID, more people actively participate in social collaboration. Therefore we did not measure for COVID.

3

Digital technology in literature review

Improved in line number 82-90

4

Contribution of the paper

Added on line number 532-535

Round 2

Reviewer 1 Report

The authors took into account the comments and significantly improved the text.

Author Response

1.      Arguments and discussion of findings coherent, balanced and compelling can be improved.

Improvement on discussion section was done in line number 485-491.

2.      The referenced can be improved.

Four Reference were added ( No. 45, 56, 57 and 79)

Reviewer 2 Report

I have read the improvements made to your article and they are correct. However, please correct those sentences and paragraphs that remain similar to your 2015 Article. There persist sentences whose wording could be interpreted as an obvious copy. E.g., in 2.4.1.(line 232ff), lines 233 to 235, and lines 241-243 are the same as in the 2015 article. There are still similar sentences left in the article, so it is necessary for you yourselves to revise, reword or delete to avoid interpretations of self-plagiarism.

Author Response

1.      Theoretical background and empirical research (if applicable) on the topic can be improved.

Improvement was made on literature review at line number, 199, 232-234, 240-241, and 261-265.

2.      Research design, questions, hypotheses and methods can be improved.

Improvement was made on line number 296-297 and 329-31.

3.      Arguments and discussion of findings coherent, balanced and compelling can be improved.

Improvement on discussion section was done in line number 485-491.

4.      The results presentation can be improved

Improvement was made on line number 398 as well as some sentences were deleted from 2nd, 4th and 5th paragraph of result section to improve result presentation.

Issue: There persist sentences whose wording could be interpreted as an obvious copy. E.g., in (line 232ff), lines 233 to 235, and lines 241-243 are the same as in the 2015 article. There are still similar sentences left in the article, so it is necessary for you yourselves to revise, reword or delete to avoid interpretations of self-plagiarism.

We have made improvement on line number 233-234 and 240-241 to avoid interpretation of self-plagiarism.

Reviewer 3 Report

Table 1 does not provide any "descriptive statistics" of the responses collected. What I suggested was to provide a summary the survey data (i.e. the collected responses), which I still find missing. Though it is not not crucial so I will leave it as a suggestion.

Line 556-57:
".. therefore it could be generalized within the country." Perhaps this is a typo and the authors mean that it could NOT be generalized.

Author Response

Dear reviewer,

We provide Table 2. Demographic characteristics. Please check the table 2.

We also change "could be" into "could not be" in Line 556-557.

Thank you very much.

With regards,

Ilsang KO

Reviewer 4 Report

Thank you to the authors for reviewing and improving the paper.

Author Response

1.      Is the content succinctly described and contextualized with respect to previous and present theoretical background and empirical research (if applicable) on the topic?

Improvement was made on literature review at line number, 199, 232-234, 240-241, and 261-265.

2.      The results presentation can be improved.

Improvement was made on line number 398 as well as some sentences were deleted from 2nd, 4th and 5th paragraph of result section to improve result presentation.

3.      The article reference can be improved.

Four Reference were added ( No. 45, 56, 57 and 79)